# Modeling and Analysis of Morphology of Injection Molding Polypropylene Parts Induced by In-Mold Annealing

**DOI:** 10.3390/polym14235245

**Published:** 2022-12-01

**Authors:** Rita Salomone, Vito Speranza, Sara Liparoti, Giuseppe Titomanlio, Roberto Pantani

**Affiliations:** Department of Industrial Engineering, University of Salerno, Via Giovanni Paolo II, 132, 84084 Fisciano, Italy

**Keywords:** spherulite kinetics, fiber kinetics, in-mold annealing, flow-induced crystallization, mold temperature

## Abstract

It is generally recognized that high-temperature treatments, namely annealing, influence the microstructure and the morphology, which, in turn, determine the mechanical properties of polymeric parts. Therefore, annealing can be adopted to control the mechanical performance of the molded parts. This work aims to assess the effect of annealing on the morphology developed in isotactic polypropylene (iPP) injection-molded parts. In particular, a two-step annealing is adopted: the polymer is injected in a mold at a high temperature (413 or 433 K), which is kept for 5 min (first annealing step); afterward, the mold temperature is cooled down at 403 K and held at that temperature for a time compatible with the crystallization half-time at that temperature (second annealing step). The characterization of morphology is carried out by optical and electronic scanning microscopy. The temperature of the first annealing step does not influence the thickness of the fibrillar skin layer; however, such a layer is thinner than that found in the molded parts obtained without any annealing steps. The second annealing step does not influence the thickness of the fibrillar skin layer. The dimension of spherulites found in the core is strongly influenced by both annealing steps: the spherulite dimensions enlarge by the effect of annealing steps. A model that considers spherulite and fibril evolutions is adopted to describe the effect of molding conditions on the final morphology distribution along the part thickness. The model, which adopts as input the thermo-mechanical histories calculated by commercial software for injection molding simulation, consistently predicts the main effects of the molding conditions on the morphology distributions.

## 1. Introduction

Injection molding is one of the preferred processes for obtaining polymeric parts with high geometry accuracy in several fields, including transportation, construction, and biomedicine. Strength and rigidity are two important properties of polymeric parts for applications in the field of structural part production. Polyolefins, even if widely adopted to produce commodities because of their low price, generally do not show sufficient strength and rigidity for such applications. In recent years, the strategies to increase the strength and rigidity of injection-molded polyolefins, especially polypropylene, are attracting relevant research interest [1]. Indeed, the optimization of the crystallization process through the control of process parameters, and the introduction of fillers and nucleating agents, can lead to an increase in polyolefin strength and rigidity up to the level required for the final applications [2,3].

Heat treatments, namely annealing, improve polymer properties by removing the residual stress in polymer products, perfecting the crystalline structure, and vanishing the crystal defects that arise during the process [4,5]. Thus, annealing can be used to tailor the polymer microstructures [6]. Generally, this technique is applied to polymers with slow crystallization in processing conditions. Boruvka et al. studied the in-mold annealing effect on the morphology, crystallization, and mechanical properties of injection-molded nanocomposites made of poly-lactic acid [7]. They found that in-mold annealing enhances tensile modulus, crystallinity, and thermal resistance. Yang et al. analyzed the influence of annealing conducted under several temperatures on the mechanical performances of a poly-lactic acid. They also found that the tensile strength can be improved without compromising the tensile ductility with a proper selection of the annealing conditions [8].

Few studies are devoted to the analysis of the effect of in-mold annealing on the morphology and crystallization behavior of polyolefins. In these cases, the analysis of the effect of annealing on morphology is even more complicated since several morphological features form during the process: fibrils oriented along with the flow direction close to the part surface and spherulites in the part core [9]. Liao et al. [10] analyzed the effect of annealing on lamellar structure in micro-injection-molded polyethylene parts. They found that annealing induced the perfecting of lamellae crystals, namely a structural improvement that led to better mechanical performances. Bai et al. [6] analyzed the effect of annealing on polypropylene with flow-oriented crystalline structures. They found a general decrease in the molecular orientation, the formation of secondary kebabs, and the perfecting of primary kebabs due to the annealing. The appearance of secondary crystallization involves the reorganization of the amorphous phase, which mostly determines an improvement of the mechanical properties [11]. Liparoti et al. conducted experiments with a short (with annealing time comparable to the packing time) in-mold annealing and found that the annealing induced an increase in the lamellar thickness, which led to the increase in the mechanical performances of polypropylene [12]. The possibility of predicting the effect of the in-mold annealing on crystallization, thus on the part performances, would widen the application of this technique to the industrial level. However, attempts to predict the formation of all the crystalline features formed during injection molding is scarce.

In this work, the injection molding process coupled with in-mold annealing was conducted on an isotactic polypropylene (iPP) to assess the influence of in-mold annealing on the morphology developed within the molded parts. A model for the description of the crystallinity evolution was proposed and applied to predict the formation of each morphological feature formed during the process. Particularly, a two-step simulation was conducted: the process evolution was simulated through commercial software for injection molding, then the processing variable evolutions obtained from the first step were used as inputs for the prediction of crystallinity evolution. This approach is expected to empower commercial software for injection molding, giving the possibility to predict the formation of several morphological features (now only isotropic structures can be described), thus, describing the overall morphology of the molded part. 

## 2. Materials and Methods

Injection molding experiments were conducted on a well-characterized iPP (T30G, Basell, Switzerland) [13,14,15].

HAAKE Minijet II by Thermo Scientific (HAAKE MARS, Milan, Italy) was adopted for producing the specimens with different morphology distributions. This machine is a mini-injection molding system that adopts a pneumatic piston to control the pressure during the molding. The molds for HAAKE Minijet II present a truncated cone shape with a diameter that changes from 50 mm (at the gate side) to 35 mm over a length of about 90 mm. The specimens were obtained with a rectangular cavity having the following dimensions: 1 mm thick, 10 mm wide, and 60 mm long.

The following conditions were adopted for the tests: 473 K as injection temperature, 200 bar as filling pressure, and 5 s as filling time. An in-mold annealing stage was also adopted to tailor the morphology distribution within molded parts. The annealing was conducted in two steps: the first step was conducted with T_mold_ temperature (see Table 1) for 5 min, and the second step was conducted at 403 K for 120 min. Figure 1 shows the temperature evolution during an in-mold annealing test carried out with two steps; the processing stages are also indicated in the figure.

The temperature selected for the second annealing step, 403 K, accounts for the half-crystallization time of the iPP adopted in this work [9] in order to allow crystallization during the second annealing step. The operating conditions of all tests carried out in this work are summarized in Table 1.

Thin slices (150 µm thick) were cut from the central part of the molded slab in the flow-thickness plane by Leica RM 2265 slit microtome (Leica Biosystem, Buccinasco, Milan, Italy). In particular, a sample was obtained by considering the first 30 mm of the molded part downstream of the injection location and then, as shown in Figure 2, slices were cut close to the half-width of such a sample. Thin slices were analyzed by optical microscope Olympus BX51 (Olympus Italia S.R.L., Segrate, Milan, Italy). The slices were etched following the procedure reported by Basset [16,17] and analyzed by scanning electron microscopy (SEM). 

Figure 2 shows the sketch of the positions for the optical microscopy.

A desktop SEM (Phenom ProX, Phenom-World BV, Netherlands) was adopted for morphological characterization. Before the analysis, the samples were coated with a thin film of gold by sputtering. 

Moldflow software (Moldflow 2018, Autodesk Inc., San Rafael, CA, US) was adopted for the simulations of the injection molding process in the conditions reported in Table 1 with the main purpose of describing shear rate and temperature distributions along the cavity thickness at several positions along the flow path. A Moldflow material database was developed according to the characterization of the iPP adopted in this work [18,19]. In particular, the ad-hoc developed database includes parameters of the quiescent crystallization model of the iPP. The dual-domain mesh adopted for the simulations is composed of 12,354 triangular elements. 

Shear rate and temperature evolutions obtained from the Moldflow simulations were used as inputs for the model proposed to predict the crystallinity evolution in the forms of spherulites and fibrils.

## 3. The Crystallization Model

The model for describing crystallization evolution is given in a previous paper [20,21,22,23]. It is briefly reported in the Appendix A. Particularly, the overall relative crystallinity degree ξ of the alpha crystalline phase, the predominant phase in the injection-molded parts made of the iPP adopted in this work, is given by Equation (1)
(1)$$\xi =\xi_{fibrils}+\xi_{spherulites}$$

Fibrils and spherulites compete for the same amorphous space; their evolutions are given by Kolmogoroff’s Equations (reported in the Appendix A). Accordingly, the space occupied by spherulites, $\varphi_{0}$, and the space occupied by fibrils, $\psi_{0}$, can be described by Equation (2).
(2a)$$\varphi_{0}=\frac{4}{3}\pi \int_{-\infty }^{t}{\dot{N}}_{s}\left(s\right){\left[\int_{s}^{t}G_{s}\left(u\right)du\right]}^{3}ds$$
(2b)$$\psi_{0}=2\pi \int_{-\infty }^{t}{\dot{N}}_{f}\left(s\right)\int_{s}^{t}G_{f}\left(u\right){\left[\int_{u}^{t}G_{s}\left(p\right)dp\right]}^{2}du\;ds$$
where ${\dot{N}}_{s}$ is the nucleation rate of the spherulites and $G_{s}$ is the radial growth rate of the spherulites. ${\dot{N}}_{f}$ represents the nucleation rate of the fibrils, and $G_{f}$ is the fibrils axial growth rate (in other words, the growth rate along the flow direction). The fibril growth rate along the direction transverse to the flow is assumed to be equal to the spherulite growth rate, $G_{s}$, under the same flow conditions. All crystalline functions are dependent upon temperature and pressure. Moreover, the effect of flow on the crystallization process was accounted for via the molecular stretch. The evolution of molecular conformation tensor A during the process was described with Maxwell’s model (as reported in the Appendix A). Under the hypothesis of simple shear flow (considered valid in the cases of injection molding in a constant thickness cavity), components $A_{xy}$ and $A_{xx}$ of the molecular stretch tensor are given by Equations (3):(3a)$$\frac{\partial A_{xy}}{\partial t}=\frac{A_{xy}}{\lambda }-\dot{\gamma }$$
(3b)$$\frac{\partial A_{xx}}{\partial t}=\frac{A_{xx}}{\lambda }+2\dot{\gamma }A_{xy}$$
where *x* is the flow direction, *y* is the direction transversal to the flow, λ is the relaxation time, and $\dot{\gamma }$ is the shear rate. Dependence upon temperature, pressure, crystallinity, and molecular stretch of the relaxation phenomenon was accounted for and the corresponding expressions are reported in the Appendix A. The molecular stretch, Δ, is given by the difference between the two main eigenvalues of the molecular stretch tensor and in simple shear, is expressed by:(4)$$\Delta =\sqrt{A_{xx}{}^{2}+4A_{xy}^{2}}$$

The experimental investigation conducted on the iPP considered in this work showed that fiber formation is largely favored with respect to the spherulitic one at a high level of molecular stretch, namely Δ. Moreover, fibers can form only when a sufficient level of molecular orientation is reached. This was considered through the introduction of a threshold value of the molecular stretch in the crystallization model.

The average spherulite radius, *R*, can be obtained from Equation (5)
(5)$$R=\sqrt[{3}]{\frac{3\;\xi_{spehrulites}}{4\pi \;N_{a}}}$$
where $\xi_{spherulites}$ and $N_{a}$ refer to the final values. When Kolmogoroff’s approach is followed, the number $N_{a}$ of active nuclei at the end of the crystallization process can be calculated as
(6)$$N_{a}=\int_{0}^{t}\frac{dN_{s}\left(p\right)}{dp}\left(1-\xi \left(p\right)\right)dp$$

## 4. Results and Discussion

Figure 3 shows the optical micrographs of the specimens cut along the flow thickness plane obtained in several injection molding conditions (see Table 1).

The morphology of all the analyzed specimens is composed of a banded-colored area, which corresponds to the fibrillar layer close to the sample surface, and a granular area in the core, corresponding to the spherulitic layer. Sample S shows the largest fibrillar layer covering most of the sample thickness. Previous results [24] on molded parts obtained adopting the same iPP, and similar molding conditions highlighted that the skin layer thickness depends on the modulation of the cavity temperature. The increase in cavity temperature from 298 to 413 K induces a significant reduction in the fibrillar layer thickness. A further increase in the cavity temperature has a negligible effect on the fibrillar layer thickness, and probably a minimum of the skin layer thickness is attained. Indeed, it was already observed [24] that the lower the cavity temperature, the thicker the skin layer. Moreover, the second annealing step does not affect the extension of the fibrillar layer. The preliminary observation conducted by optical microscopy seems to suggest that the fibrillar layer forms during the early stage of the process, i.e., during the filling.

Figure 4 shows optical and SEM micrographs for samples S, 413_FA, and 413_SA, 10 mm from the gate. SEM images captured in several positions along the sample’s half-thickness confirm the morphology distribution already observed by optical micrographs.

Close to the sample surface, SEM micrographs acquired in the banded-colored area of the optical micrograph are characterized by tightly packed fibrillar structures. In the transition zone, SEM micrographs show intermediate morphology at the boundary between the fibrillar layer and the spherulitic one. Nuclei, which form at the boundary, cannot grow toward the surface due to fibrils that hinder growth along the direction transversal to the flow. Such nuclei grow only toward the midplane, where the crystallization in the form of spherulites is favored by lower values of shear rate (as discussed below). Therefore, the final morphology in the transition zone is characterized by the presence of not fully developed spherulites with dimensions smaller than those observed for spherulites in the sample core. This finding is consistent with the higher value of nucleation induced by an almost strong flow experienced by the polymer in the transition zone. Close to the sample mid-plane, SEM micrographs acquired in the granular area of the optical micrographs confirm the presence of spherulites. The adopted molding conditions influence the dimension of spherulites. Spherulites significantly enlarge when a second step is carried out. Further, for tests conducted with 433 K mold temperature, similar increases in the spherulite dimensions with respect to the test S were observed. The experimental findings suggest that if the crystallization in the inner part of the specimen occurs during annealing steps, it occurs within high temperatures, at which the nucleation is slower; thus, the crystals are allowed to grow for a higher extent before impingement with respect to case S.

### Morphology Prediction

Numerical simulations with Moldflow for the cases reported in Table 1 were carried out. First, temperature evolutions obtained by Moldflow simulations at the cavity-melt interface have been compared with the evolutions recorded during the process at the same distance from the gate. Figure 5 shows the comparisons between the calculated and recorded temperature evolutions for cases S and 413_FA at the cavity surface, 10 mm downstream from the gate.

The temperature is high at the first contact of the melt with the cavity surface; after that, it decreases down to the temperature of the mold. In case S, the temperature decreases fast toward 298 K; in case 413_FA, the temperature decreases toward 413 K since the whole mold is kept at this temperature for 5 min. At the end of annealing, the temperature decreases toward 298 K. Simulations consistently reproduce the main features of the experimental temperature evolutions. Particularly, the temperature decrease from the melt temperature down to the cavity surface temperature (temperature decrease rate depends on the adopted conditions) is consistently reproduced by the simulation.

Secondly, morphology distributions within the samples were predicted with the model for describing crystallization and compared with optical micrographs. To this aim, Moldflow temperature and shear rate evolutions were adopted as input for the crystallization model. As already described, the crystallization model couples the model for the description of the molecular stretch to the model for the description of crystallinity evolution, allowing morphology distribution prediction.

Figure 6 shows temperature, shear rate, molecular stretch, and crystallinity evolutions in two positions along the thickness, one close to the mold surface and the other toward the core, both for the position 10 mm from the gate for case S. The thickness is normalized on the cavity dimension, and it ranges from −0.5 to 0.5 (0 is the cavity midplane).

The morphology is determined by the simultaneous local evolutions of temperature, flow rate, and relaxation time, see Equation (3a,b). Close to the cavity surface (Figure 6a,c), at the early stage of the process, the shear rate is very high; as time proceeds, the local temperature decreases toward the surface temperature, and, under standard conditions, this determines an increase in the relaxation times. Thus, a high level of molecular stretch Δ survives, and crystallization proceeds in the form of fibrils. The final molecular stretch usually decreases with the distance from the mold wall, but it can also undergo some local increases, as shown by the evolution of Δ reported in Figure 6d. Indeed, as cooling proceeds, after the initial filling peak, the orientation decreases because relaxation phenomena actively counteract the effect of flow. The effect of the subsequent packing flow overcomes the local stretch relaxation, and Δ locally increases again. As the packing flow undergoes a significant decrease, stretch Δ also decreases, as shown in Figure 6d. When material solidification is completed, at about 2.5 s, Δ remains constant and frozen in the part. Thus, the low level of molecular stretch Δ favors the formation of spherulites.

Figure 7 shows the temperature, shear rate, molecular stretch, and crystallinity evolutions calculated by the model on the basis of the Moldflow results in two positions along the thickness, one close to the mold surface and the other toward the core for case 413_FA.

Figure 7a,b show that, at both positions, the temperature and shear rate are both high at the beginning of the process when the cavity filling occurs. The high temperatures held for 300 s favor material flow for times (0.5 s which corresponds to the cavity filling) longer than the ones observed in case S (see Figure 6a,b). After filling, flow strength decreases, consequently, the shear rates assume values smaller than two orders of magnitude (and becomes of the order of 1 s^−1^) and become zero at about 9 s. The slope change in the shear rate evolutions at 5.5 s corresponds to the pressure release at the end of the holding stage. At that time, the temperature remains constant for the vanishing of the hot convective term. The shear rate follows the same trend at both positions considered along the thickness; obviously, the values close to the cavity surface are significantly larger than the values calculated toward the midplane. Concerning temperature, it decreases within a higher temperature range with respect to case S due to the high mold temperature adopted. Figure 7c shows that the molecular stretch is high (about 20) up to 0.5 s; after that, it has halved due to the shear rate decrease in the presence of high temperatures (which favors molecular relaxation). The high values of the molecular stretch favor the crystallization in the form of fibrils that, in turn, avoid relaxation phenomena. Consequently, even in the presence of a weaker flow, the molecular stretch starts to increase again at 0.5 s. That increase stops since the shear rate becomes sufficiently small (at about 6 s), and the molecular stretch assumes constant and high values. Figure 7d shows the molecular stretch evolution calculated close to the midplane. In that position, due to high temperatures and low shear rates, molecular stretch never achieves values sufficient to induce fibrillar crystallization. There, crystallization occurs under quiescent conditions during mold cooling (see the inset of Figure 7b). The crystallinity evolutions in the form of spherulites evaluated by Moldflow simulations is also reported in Figure 7d. It can be observed that the crystallinity prediction obtained by Moldflow simulations (blue line) and the results of the model proposed in this work (green line) are consistent.

Figure 8 shows the temperature, shear rate, molecular stretch, and crystallinity evolutions calculated by the model on the basis of the Moldflow results in two positions along the thickness, one close to the mold surface and the other toward the core for case 433_FA.

Temperature and shear rate evolutions follow the same trend already discussed for case 413_FA. However, the temperature remains within higher values for longer times. Fibers form only at the sample surface, namely at a normalized thickness of 0.5 (Figure 8a,c). For all the remaining positions along the cavity thickness, the temperature ranges determine the amount of molecular stretch achieved during the process: since, for case 433_FA, temperatures are high and molecular relaxation is favored. Therefore, molecular stretch values never reach levels sufficient to allow fibril formation in any position along the cavity thickness, unlike the sample surface. Crystallization in the form of spherulites occurs later in the process, during the cooling toward the ambient temperature. Further, in this case, the crystallinity evolutions obtained by Moldflow simulations overlap those obtained by the model. A long annealing time allows a homogeneous temperature distribution along the sample thickness and almost simultaneous crystallization of the polymer along the sample thickness.

Figure 9 shows the crystallinity degree as a function of temperature experienced by the polymer close to the midplane for cases S, 413_FA, and 433_FA.

In condition S, crystallization occurs at low temperatures, within a temperature range of 360–315 K, due to the high cooling rate experienced by the polymer during the process. When a single annealing step is adopted (cases 413_FA and 433_FA), the two crystallinity evolutions overlap. The crystallization occurs during the cooling of the whole mold toward the ambient temperature. The cooling rate experienced by the polymer is lower, and crystallization occurs at a higher temperature with respect to case S, within a narrower range, 380–360 K. 

Figure 10 shows the temperature and crystallinity evolutions for cases 413_SA and 433_SA at 0.06 normalized distance from the midplane. In Figure 10b, the crystallinity predictions obtained by Moldflow simulations for both cases are shown.

In both cases, crystallization occurs during the annealing step at 403 K when the temperature is constant along the whole cavity thickness and under quiescent conditions. Moreover, Moldflow simulations consistently predict crystallinity evolutions. In case 413_SA, the lower temperatures experienced by the polymer during the first annealing stage induced faster nucleation and growth of the spherulitic structures with respect to case 433_SA; thus, crystallization occurs earlier than in case 433_SA.

On the basis of the considerations reported above on the crystallinity evolution and its dependence on the molecular stretch, the predicted morphology distribution along the cavity thickness for all the cases examined in this work was compared with experimental observations. Figure 11 shows, for each case reported in Table 1, the predicted distributions of fibrils and spherulites. Optical micrographs are also reported for each case.

Figure 11 shows that, for each condition considered in this work, the total content of the α-phase (obtained by summing up fibers and spherulite contents) is constant along the normalized distance from the midplane. This is consistent with previous experimental results [12] that evidenced an almost constant distribution of the total crystallinity degree with a predominant content of the α-phase along the thickness of the iPP molded sample.

Figure 11a refers to case S and shows that the simulation almost correctly predicts the final distribution of fibrils and spherulites along the cavity thickness. Dependencies of the final fibrillar layer thickness upon the molding conditions are correctly described by the simulation. Particularly, the fibrillar layer thickness decreases as the mold temperature increases and remains unchanged when a second annealing step at 403 K is considered. However, for a 433 K mold temperature, the formation of fibers is predicted only at the sample surface.

An increase in the fibrillar kinetics at high temperatures would favor the fibrillar layer formation when a 433 K mold temperature is adopted. Such an effect were also observed in the experimental investigation conducted on the iPP fibrillar kinetics [21]. Moreover, the accounting of the interplay between crystallinity and viscosity would induce, as the fibrillar layer forms, an increase in the thickness of the solidified layer close to the sample surface. This, in turn, would increase the flow strength that favors the fibril formation; consequently, the fibrillar layer enlarges, allowing for the recovery of the discrepancies between the experimental observations and the model predictions. This aspect is more relevant for the cases with a 433 K mold temperature where the material solidification, namely fibril formation, is predicted at high temperatures.

Table 2 compares the spherulite-averaged radii evaluated from the SEM micrographs with spherulite radii predicted, adopting the model reported above and predicted by Moldflow simulations. 

Predictions of radii are in the same order of magnitude as the experimentally measured radii. Samples 413_FA and 433_FA show the same average radii, which are larger than the S case. The second annealing step at 403 K leads to a significant increase in the spherulite dimensions: cases 413_SA and 433_SA show similar spherulite dimensions. The predictions almost correctly describe the dependencies upon the molding conditions of the spherulite’s dimensions. However, the dimensions of spherulites are underestimated in the standard case and in the cases with a single annealing step. The spherulite’s crystalline functions were defined by extrapolating the results of the experimental investigation conducted in conditions far from the ones experienced by the polymer during the process. For the case with a second annealing step at 403 K, crystallization occurs under isothermal and quiescent conditions similar to those adopted during the experimental investigation on crystallization phenomenon [22], whereas crystallization occurs during cooling toward the mold temperature for all the other cases. Therefore, the model’s predictions of the final dimensions of the morphological features are more accurate for second annealing cases with respect to the other cases.

## 5. Conclusions

This work analyzes the effect of processing conditions, concerning mold temperature and in-mold annealing, on the morphology developed within the molded parts. Molded parts were characterized by optical and electronic microscopy: they show a fibrillar layer close to the cavity surface and a spherulitic core. It was found that morphology was strongly affected by processing conditions: fibrillar layer reduced by applying in-mold annealing, due to the increase of the mold temperature; the second annealing step had a negligible influence on the fibrillar layer extension. Furthermore, the thickness of the fibrillar layer is scarcely influenced by the mold temperature adopted during the in-mold annealing. Spherulites’ dimensions enlarged due to the effect of both the mold temperature and annealing steps. 

To predict the morphology distributions induced by the process, a model that couples the description of the fibrils and spherulite evolution with the description of the molecular stretch evolution was adopted. The injection-molding tests in standard conditions and in presence of in-mold annealing were simulated by Moldflow. Predicted temperature and shear rate evolutions were used as input to the proposed model for predicting morphology distributions. The model was able to describe the formation of fibrils close to the sample surface and spherulites in the core of the sample. The main effects of the molding conditions on the morphology distributions were effectively predicted by the model. In the presence of in-mold annealing, a decrease in the fibrillar layer thickness with respect to the standard case was predicted. Further, the predicted fibrillar layer formed during the early stage of the process; thus, it is not influenced by the second annealing step. However, the effect of the mold temperature in the range of considered temperatures was overestimated by the predictions. The model almost accurately described the increase in the spherulites’ dimensions induced by the in-mold annealing. In particular, the second annealing step had a significant effect on the spherulites’ dimensions. The experimentally observed and predicted spherulites’ dimensions were of the same order of magnitude for all considered cases. 

It is worth pointing out that, with the proposed approach, an acceptable description of the final morphologies (fibrils and spherulites) distribution in molded parts can also be obtained by adopting simulations conducted with commercial software that does not consider the fibril content at all. However, a more accurate prediction of the morphology distributions could be achieved including the models of the different morphology evolution in simulation software and considering the effects of crystalline structures on the polymer’s properties (viscosity, relaxation time).

## Figures and Tables

**Figure 1 polymers-14-05245-f001:**
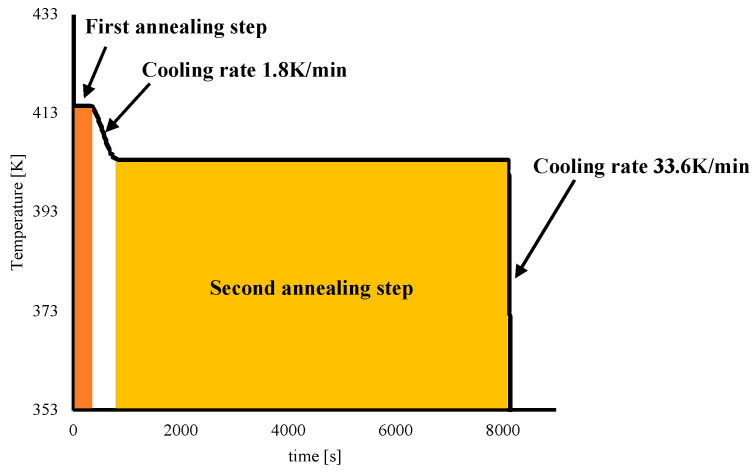
Temperature evolution recorded during the experiment with double-step annealing.

**Figure 2 polymers-14-05245-f002:**
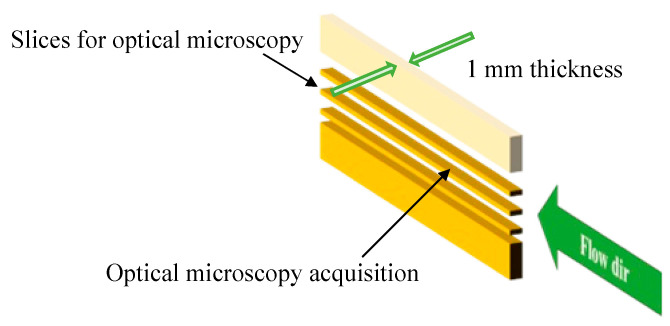
Sketch showing the position inside the injected sample adopted for the optical microscopy.

**Figure 3 polymers-14-05245-f003:**
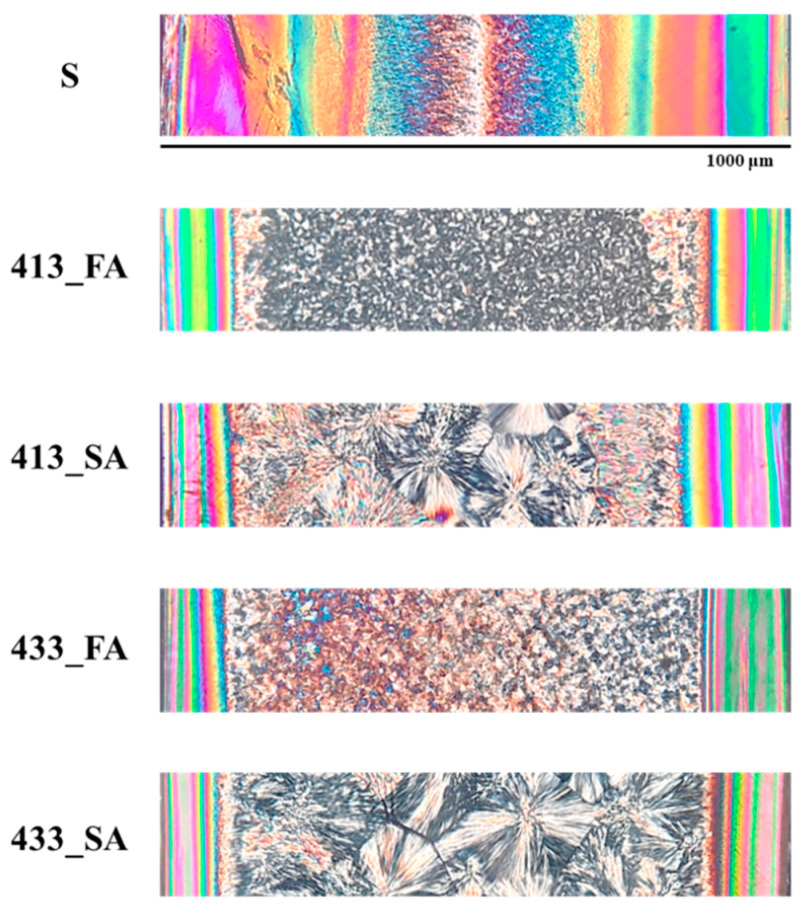
Optical micrographs of the specimens cut along the flow thickness plane 10 mm from the gate. The specimens were obtained following the conditions reported in Table 1.

**Figure 4 polymers-14-05245-f004:**
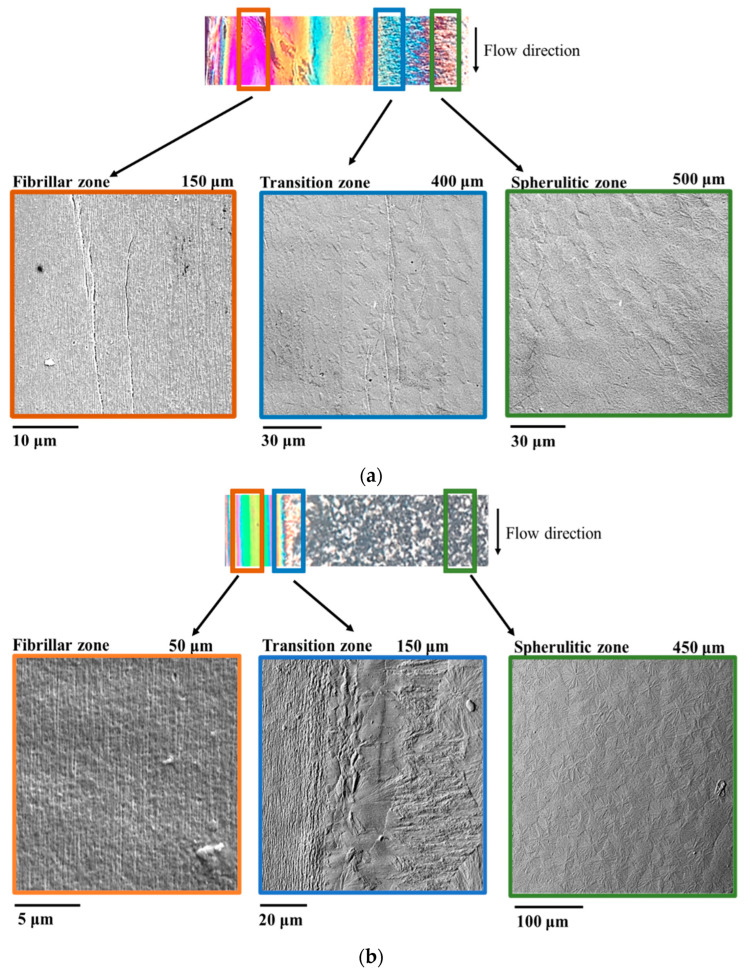
Optical and SEM micrographs for samples S (**a**), 413_FA (**b**), and 413_SA (**c**) in several positions along the half thickness 10 mm from the gate. For each micrograph, the distance from the sample surface is also indicated.

**Figure 5 polymers-14-05245-f005:**
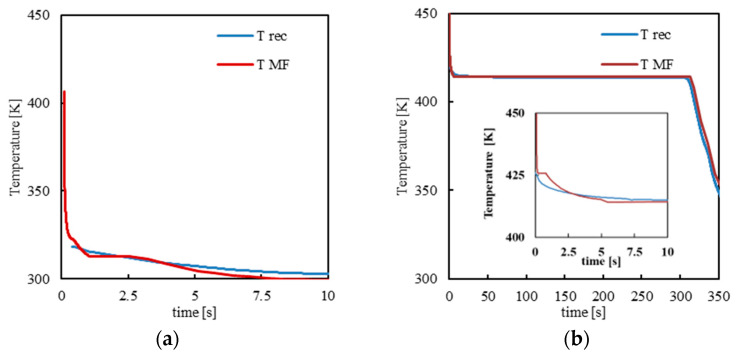
Temperature evolution for cases S (**a**) and 413_FA (**b**) as recorded during the injection molding test (T_rec_) and calculated by Moldflow simulations (T_MF_) at the cavity surface.

**Figure 6 polymers-14-05245-f006:**
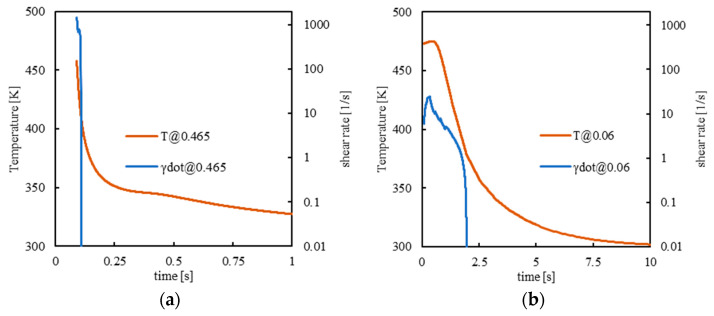
Temperature, shear rate, molecular stretch, and crystallinity evolutions for case S at 0.465 (**a**–**c**) and 0.06 (**b**–**d**) normalized distances from the midplane.

**Figure 7 polymers-14-05245-f007:**
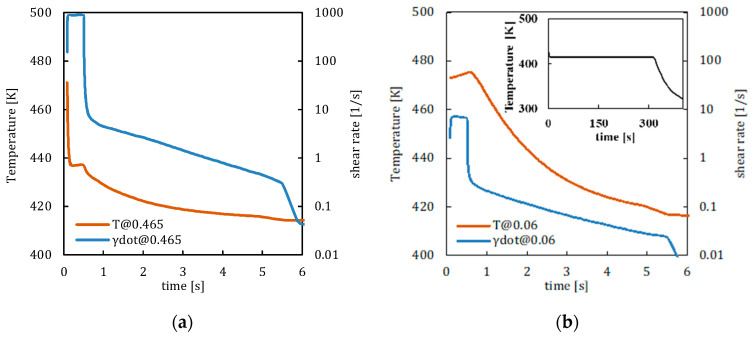
Temperature, shear rate, molecular stretch, and crystallinity evolutions for case 413_FA at 0.465 (**a**–**c**) and 0.06 (**b**–**d**) normalized distances from the midplane 10 mm from the gate. Crystallinity predictions calculated by Moldflow are also shown in (**d**) as the blue line.

**Figure 8 polymers-14-05245-f008:**
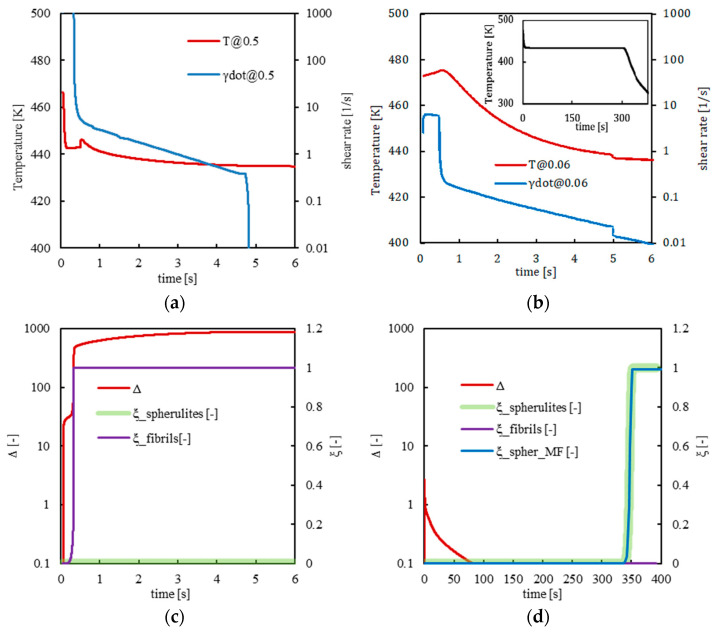
Temperature, shear rate, molecular stretch, and crystallinity evolutions for case 433_FA at 0.5 (**a**–**c**) and 0.06 (**b**–**d**) normalized distances from the midplane at 10 mm from the gate. Crystallinity predictions calculated by Moldflow are also shown in (**d**) as the blue line.

**Figure 9 polymers-14-05245-f009:**
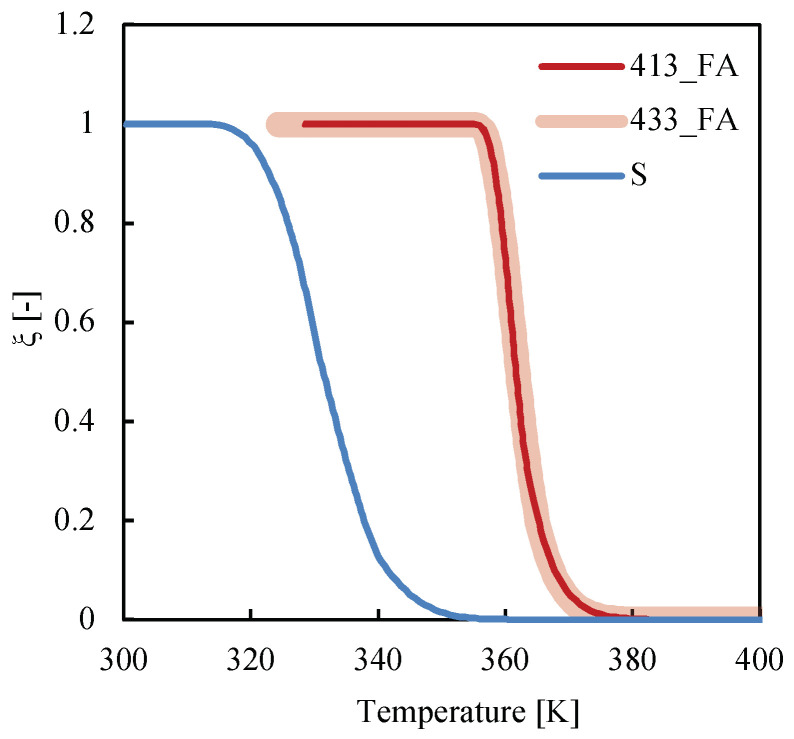
Crystallinity evolution with temperature for cases S, 413_FA, and 433_FA at 0.06 normalized distance from the midplane 10 mm from the gate.

**Figure 10 polymers-14-05245-f010:**
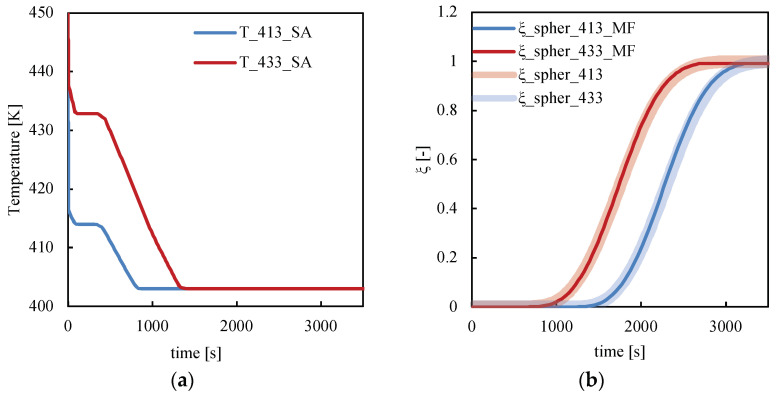
Comparison between (**a**) temperature and (**b**) crystallinity evolutions calculated by Moldflow simulations and by the model at 0.06 normalized distance from the midplane for the cases 413_SA and 433_SA.

**Figure 11 polymers-14-05245-f011:**
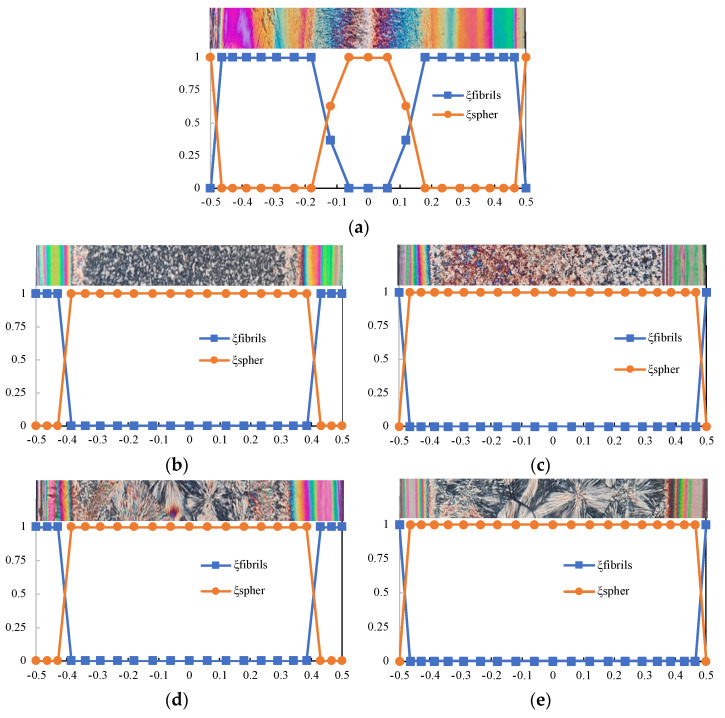
Distributions of spherulites and fibrils along the normalized distance from the midplane for each condition reported in Table 1 (**a**) S; (**b**) 413_FA; (**c**) 433_FA; (**d**) 413_SA; and (**e**) 433_SA.

**Table 1 polymers-14-05245-t001:** Injection molding conditions (T_mold_: cavity temperature, t_fa_: first annealing time, T_sa_ second annealing temperature, t_sa_: second annealing time).

Sample Name	T_mold_ (K)	t_fa_ (min)	T_sa_ (K)	t_sa_ (min)
S	298	-	-	-
413_FA	413	5	-	
413_SA	413	5	403	120
433_FA	433	5	-	-
433_SA	433	5	403	120

**Table 2 polymers-14-05245-t002:** Spherulitic average radii measured from SEM analyses (R_exp_), obtained from Moldflow simulations (R_MF_), and radii calculated adopting the model reported in this work (R_calc_).

Case	R_exp_ (μm)	R_MF_ (μm)	R_calc_ (μm)
S	7.5 ± 2	1	2
413_FA	17 ± 2	7	11
433_FA	17 ± 2	7	10
413_SA	70 ± 5	74	90
433_SA	75 ± 10	74	90

## Data Availability

The data that support the findings of this study are available from the corresponding author upon reasonable request.

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
