# Peer review of "Modeling and Analysis of Morphology of Injection Molding Polypropylene Parts Induced by In-Mold Annealing"

_polymers, 2022, doi:10.3390/polym14235245_

Round 1

Reviewer 1 Report

“Modeling and analysis of morphology of injection molding     polypropylene parts induced by in-mold annealing” investigates relationship between the annealing procedure in the injection process and the morphology of the injected samples and test a proposed model that explain experimental results.

Revision or comment

The work is in the very broad and interesting field of the injection molding process. The aim to tie the production process technique to the morphology of the injected samples and hence to the capability to influence the thermomechanical properties of the produced parts could be useful in several application fields.

However, some details can be added for a better comprehension of the research activity.

The annealing temperature does not affect the skin layer but without it skin layer is much thicker. Is there a minimum annealing temperature that could produce a thinner skin layer or is more important the duration of the annealing? The annealing step is time and energy consuming for a broad production.

How were slices cut from the sample? Even if details are presents in the references, a brief note could be useful to the reader.

How many samples were considered to obtain the slices for each process conditions?

Author Response

The authors thank the reviewer for the time and efforts devoted to the revision of the manuscript. The answers to the reviewer’s suggestions are given point by point in the attached file.

Best regards,

Vito Speranza

Reviewer 2 Report

The plastic injection process is one of the most common methods of forming products. Thanks to the appropriate parameters of the machine and the mold, we can influence the quality of the product. The authors presented the problem related to the mold temperature, the residence time of the material in the mold and the formation of the appropriate structure. The residence time and temperature have a significant impact on the formation and expansion of the crystalline phase in partially crystalline polymers. The development of crystals has an impact on the properties of the finished product and we can optimize these properties through parameters. Looking critically at the residence time in the mold and its temperature, the total cycle time will increase significantly, thereby ultimately increasing the manufacturing cost of a particular batch of products. The work lacks DSC curves on the basis of which the degree of crystallization can be determined. They would be a perfect complement. The adopted research methodology is correct. Authors should choose the same temperature units. The analyzed literature is up-to-date.

Author Response

(The authors gave the same response as above.)

Round 2

Reviewer 2 Report

The changes made are accepted. They supplement the information on the purpose of the conducted research and the results obtained. Of course, there are still a few minor shortcomings that could be improved, but I accept the corrections as they stand.